# Comparing Perioperative Outcome Measures of the Dynamic Hip Screw and the Femoral Neck System

**DOI:** 10.3390/medicina58030352

**Published:** 2022-02-26

**Authors:** Marcel Niemann, Karl F. Braun, Sufian S. Ahmad, Ulrich Stöckle, Sven Märdian, Frank Graef

**Affiliations:** 1Center for Musculoskeletal Surgery, Charité—Universitätsmedizin Berlin, Corporate Member of Freie Universität Berlin, Humboldt-Universität zu Berlin, Berlin Institute of Health, 13353 Berlin, Germany; karl.braun@charite.de (K.F.B.); sufian@ahmadortho.com (S.S.A.); ulrich.stoeckle@charite.de (U.S.); sven.maerdian@charite.de (S.M.); frank.graef@charite.de (F.G.); 2Julius Wolff Institute for Biomechanics and Musculoskeletal Regeneration, Charité—Universitätsmedizin Berlin, 13353 Berlin, Germany; 3Department of Trauma Surgery, University Hospital Rechts der Isar, Technical University of Munich, 81675 Munich, Germany; 4Department of Orthopedic Surgery, Hannover Medical School, 30625 Hannover, Germany

**Keywords:** dynamic hip screw, femoral neck system, femoral neck fracture, individual medicine, minimal-invasive surgery, multiple trauma, geriatrics

## Abstract

*Background and Objective*: Various fixation devices and surgical techniques are available for the management of proximal femur fractures. Recently, the femoral neck system (FNS) was introduced, and was promoted on the basis of less invasiveness, shorter operating time, and less fluoroscopy time compared to previous systems. The aim of this study was to compare two systems for the internal fixation of femoral neck fractures (FNF), namely the dynamic hip screw (DHS) with an anti-rotation screw (ARS) and an FNS. The outcome measures included operating room time (ORT), dose–area product (DAP), length of stay (LOS), perioperative changes in haemoglobin concentrations, and transfusion rate. *Materials and Methods*: A retrospective single-centre study was conducted. Patients treated for FNF between 1 January 2020 and 30 September 2021 were included, provided that they had undergone closed reduction and internal fixation. We measured the centrum-collum-diaphyseal (CCD) and the Pauwels angle preoperatively and one week postoperatively. *Results*: In total, 31 patients (16 females), with a mean age of 62.81 ± 15.05 years, were included. Fracture complexity assessed by the Pauwels and Garden classification did not differ between groups preoperatively. Nonetheless, the ORT (54 ± 26.1 min vs. 91.68 ± 23.96 min, *p* < 0.01) and DAP (721 ± 270.6 cGycm² vs. 1604 ± 1178 cGycm², *p* = 0.03) were significantly lower in the FNS group. The pre- and postoperative CCD and Pauwels angles did not differ statistically between groups. Perioperative haemoglobin concentration changes (–1.77 ± 1.19 g/dl vs. –1.74 ± 1.37 g/dl) and LOS (8 ± 5.27 days vs. 7.35 ± 3.43 days) were not statistically different. *Conclusions*: In this cohort, the ORT and DAP were almost halved in the patient group treated with FNS. This may confer a reduction in secondary risks related to surgery.

## 1. Introduction

Femoral neck fractures (FNF) have an enormous socioeconomic impact on modern society. The total number of hip fractures is expected to increase from 1.26 million in 1990 to 21.3 million by 2050 [1]. These fractures have been reported to negatively impact patients’ functional status, quality of life, and independence [2]. Furthermore, fractures close to the hip are strongly associated with a pronounced risk of cardiovascular complications and mortality [3].

Several authors have worked on standardised treatment concept that take into account the fracture location, fracture classification, and patients’ individual risk factors [4,5]. However, these concepts remain highly heterogeneous, especially regarding the indication for osteosynthetic reconstruction or replacement with a hemi- or total hip arthroplasty. Reconstruction is reserved for cases in which the perfusion of the femoral head is presumably not compromised. Therefore, broadly accepted fracture classification systems, such as the Pauwels [6] or Garden classification [7], help clinicians through the decision-making process. Accordingly, Pauwels type I and II and Garden type I and II fractures usually qualify for reconstruction.

Various implants for the reconstruction of an FNF are currently available. The dynamic hip screw (DHS) is the most commonly used system (Figure 1). When considering the implantation of a DHS, an additional anti-rotational screw (ARS) should explicitly be used in FNF to increase rotational stability. This combination has been associated with significantly improved traction and compression distribution on fractures [8], potentially facilitating a healing outcome. However, a recent meta-analysis observed no superiority regarding mortality, fracture consolidation rate, and revision rate when comparing the DHS to cannulated screws [9].

Recently, a new and innovative reconstruction system was introduced: the femoral neck system (FNS) (DePuy Synthes, Raynham, MA, USA) (Figure 2) [10,11,12]. This system is exclusively designed to stabilise FNF. It allows for dynamic fixation of the femoral neck, rotational stability through a screw-in-screw concept, and increased strength at the shaft due to a locking screw. Thereby, it combines the biomechanical advantages of different well-known osteosynthesis principles. Furthermore, the FNS can be applied percutaneously while maintaining the beneficial characteristics of the DHS. Biomechanical studies have shown that the FNS is as a valid alternative to the DHS with ARS and is superior to cannulated screws for the management of Pauwels type III fractures [13]. Recent clinical studies have shown that reconstructions using the FNS lead to satisfactory perioperative and clinical outcome measures [14,15,16,17,18]. To date, only one group of authors has compared the FNS with the DHS for Garden type I and II fractures in elderly patients [17]. They observed a shorter operating room time (ORT) in the FNS group, but there were no differences in the transfusion rate, local complications, length of stay (LOS), or mortality between groups. However, only including elderly patients and Garden type I and II fractures may impair study data quality and limit the implications for other clinicians.

Therefore, this study aimed to compare all patients that were stabilised using either a DHS with ARS or FNS at our institution. Particular emphasis was given to the ORT, which was our primary outcome measure. Secondary outcome measures were dose–area product (DAP), LOS, change in haemoglobin concentrations, and transfusion rate.

## 2. Methods

We conducted a retrospective study examining all FNF patients being treated at our level 1 trauma centre between January 2020 and September 2021. Approval of the local institutional review board (application number EA4/141/21) was obtained before initiation of the study. Patient data (age, gender, the American Society of Anesthesiologists [ASA] physical status classification system, Charlson Comorbidity Index (CCI) [19], trauma mechanism, fracture type according to Pauwels and Garden, LOS, and complications following surgery) were extracted from the electronic medical data system, SAP (SAP ERP 6.0 EHP4, SAP AG, Walldorf, Germany). Furthermore, perioperative data were noted including time to surgery (TTS) (including patient positioning and closed fracture reduction), ORT, DAP, transfusion rate, perioperative volume therapy, and haemoglobin concentrations prior to and following surgery.

We assessed Pauwels and centrum–collum–diaphyseal (CCD) angles in pre- and postoperative plain anterior–posterior radiographs of the pelvis using MERLIN Diagnostic Workcenter (MERLIN Diagnostic Workcenter for Microsoft Windows, Version 5.8.1, Phönix-PACS GmbH, Freiburg im Breisgau, Germany). This is displayed in Figure 3.

Statistical analysis was performed using GraphPad Prism (GraphPad Prism 9 for macOS, Version 9.3.1 [350], GraphPad Holdings, LLC, San Diego, CA, USA). Data distribution was tested using histograms and Q–Q plots. The Mann–Whitney *U* test was used for discrete and continuous variables and Fisher’s exact test was used for categorical variables. We performed outlier detection using the ROUT method with Q = 0.1% [20]. Unless otherwise stated, discrete and continuous variables are represented as the mean ± SD (95% CI), and categorical variables are presented as frequencies (%). All *p*-values are two tailed, and *p*-values < 0.05 were considered statistically significant.

## 3. Results

### 3.1. Demographics

Between January 2020 and September 2021, 31 patients (16 female) were operated on due to an FNF. Of these, 19 patients received a DHS with ARS and 12 patients received an FNS. In each group, two patients received an in situ fixation as the smallest possible intervention due to their individual perioperative risk constellations.

The mean age of the cohort was 62.81 ± 15.05 years (95% CI 57.28–68.33). Twenty-three patients (74.19%) had a low impact trauma (fall from standing height), four patients (12.9%) had a bicycle accident, two patients (6.45%) had a motorised scooter accident, one patient (3.23%) had an inline skate accident, and one patient (3.23%) had a car accident. A detailed overview of the study cohort is given in Table 1. We did not detect any significant differences between groups regarding the baseline characteristics. Especially, pre- and postoperative Pauwels classification and CCD angles did not differ between groups.

### 3.2. Outcome Measures

The ORT significantly differed (*U* = 24.5, *p* < 0.01) between the DHS group (91.68 ± 23.96 min, 95% CI 80.14–103.23) and the FNS group (54 ± 26.1 min, 95% CI 37.42–70.58). No outliers were detected. 

The DAP was 1604.19 ± 1178.16 cGycm^2^ (95% CI 1036.34–2172.04) in the DHS group and 721 ± 270.65 cGycm^2^ (95% CI 527.39–914.61) in the FNS group. Analysis revealed a significant difference between groups (*U* = 47, *p* = 0.03). One outlier was identified in the FNS group (DAP of 5407.25 cGycm^2^) and was excluded prior to analysis.

Haemoglobin concentration changes were highly comparable between the DHS group (−1.74 ± 1.37 mg/dL, 95% CI −2.42–−1.06) and the FNS group (−1.77 ± 1.19 mg/dL, 95% CI −2.52–−1.01) (*U* = 104.5, *p* = 0.89). No outliers were detected.

The LOS was 7.35 ± 3.43 days (95% CI 5.59–9.12) in the DHS group and 8 ± 5.27 days (95% CI 4.65–11.35) in the FNS group. The differences between groups were not significant (*U* = 100, *p* = 0.94). Two outliers were identified in the DHS group (LOS of 26 and 43 days) and were excluded prior to analysis.

Figure 4 shows the assessed outcome measures.

## 4. Discussion

This study represents a comparative outcome analysis of two minimally invasive fixation systems used for the surgical management of FNF, namely the DHS with ARS and the FNS. This is the first study to employ broad inclusion criteria, as we assessed Garden type I to IV fractures and did not exclude any patients due to their pre-existing medical conditions. Compared to the DHS with ARS, perioperative outcome measures revealed a shorter ORT and lower DAP when using the FNS. There were no further differences between groups regarding the assessed outcomes. Particularly, there were no inter-group differences in the pre- and postoperative Pauwels and CCD angles between groups.

Fractures to the neck of the femur represent a relevant entity of the orthopaedic surgical spectrum [1]. Frequently, these injuries result in a life-changing event for patients, especially in geriatric cohorts [2]. Therefore, therapy concepts need to be highly efficient and straightforward to prevent adverse events [3] and to continuously improve the functional and patient-reported outcomes.

There is still an ongoing debate among orthopaedic specialists about whether patients may be eligible for reconstruction instead of an arthroplasty procedure. Various individual factors need to be considered, including the specific type of fracture and individual patient characteristics such as biological age, comorbidities, and previous mobility [4]. Furthermore, the typical complications of each of these approaches also need to be taken into account [21,22]. When considering DHS, infection rates of 1.3% have been reported [23].

Reconstruction is accepted in presumably intact femoral head perfusion and in biologically young patients. Non-displaced fractures in high-risk patients with multiple comorbidities also represent well-accepted indications for fixation. Since its introduction in 2018, the FNS has expanded the spectrum of available fixation systems [12,13,14,15,16,17,18,22,24]. It is assumed to be less invasive, thereby potentially reducing perioperative risks [10,11]. Published biomechanical data for the FNS demonstrate superiority compared to cannulated screws in Pauwels type III fractures [13]. Other studies have shown that the FNS might be more resistant to varus deformation, which is one of the main failure modes of femoral neck fixation [25].

However, there is still a lack of clinical outcome data for the FNS. Stassen et al. reported data with a maximum follow-up of one year after FNS implantation [24]. The authors included all FNF types. Multiple injured and patients with severe chronic medical conditions were excluded. The authors observed an ORT of 34 ± 9.4 min, incision sizes of 45.3 ± 8.8 mm, and an LOS of 4 ± 2.8 days. These data are in concordance with our results and corroborate the assumed less invasiveness of the FNS.

Other studies compared the FNS with three cannulated screws [14,15,16,18] and observed heterogeneous outcomes. He et al. reported shorter but not significantly different ORT, less radiation, a lower complication rate, and no differences in LOS [14]. Tang et al. confirmed the reduced fluoroscopy time in the FNS group [16]. However, the authors did not observe any significant differences in ORT, blood loss, incision size, or LOS. In contrast to these reports, Hu et al. and Zhou et al. reported longer ORT and higher blood loss when the FNS was used [15,18]. Furthermore, the LOS tended to be shorter, and patients had less pain and a shorter time to walk without crutches in the FNS group [18]. When discussing these data, one must consider that the latter two studies excluded typical patients: Hu et al. solely included patients under 60 years old and Zhou et al. excluded severely ill patients and patients with pre-existing severe cognitive dysfunction [15,18]. Hence, these studies may not reflect the typical, rising elderly patient cohort [4]. Partly, our results are in line with those of the aforementioned authors. We also observed reduced ORT and DAP, but our data do not allow for an adequate comparison of the previously reported blood loss reduction. At our clinic, the total amount of intraoperative blood loss was not systematically documented in the electronic medical data system. Therefore, we assessed differences in haemoglobin concentrations following surgery and perioperative volume management in order to take dilution into account. Here, we did not find any differences between groups. This may suggest that there was no significant difference in blood loss between groups, since haemoglobin differences and perioperatively administered fluids were not different and neither group needed a transfusion prior to hospital discharge. A reason for this could be that the total blood loss is dominated by the blood loss due to the initial trauma, making the additional blood loss due to the surgical procedure, either by DHS with ARS or by FNS, relatively minor.

To date, Vazquez et al. are the only authors that have compared the DHS, the FNS, and cannulated screws [17]. However, the authors only included Garden type I and II fractures in an elderly cohort (mean age, 84.9 years). While the ORT was significantly shorter in the FNS group, there were no statistically significant differences in the transfusion rate and LOS between groups. The absolute values of the ORT and LOS compare well to our results. Furthermore, we did not observe different transfusion rates since no patient needed transfusion.

In particular, the broadly observed decrease in the ORT is of utmost importance since published data show that prolonged ORT is associated with an increased risk of postoperative complications [26]. However, the most frequent surgical complication following the osteosynthesis of the FNF is the shortening of the femoral neck and the development of avascular necroses (AVN), which is observed in up to 20% of cases [27,28]. Accordingly, the data showed a conversion rate to arthroplasty in up to 10% of cases after osteosynthesis of the femoral neck [14,17,22]. Therefore, pre-existing comorbidities, such as osteoarthrosis of the hip, severe osteoporosis, rheumatoid arthritis, and chronic kidney disease, should be taken into account, as they confer a high risk of secondary osteosynthesis failure [29]. However, we did not observe any of these complications during the primary hospital stay, which was the focus of this report.

The current study has some limitations. First, the study groups were rather small, thereby potentially limiting the statistical power. This needs to be addressed in future studies through larger cohorts. However, published clinical data for the FNS are, thus far, rare, and we provide perioperative clinical data comparing the FNS with a commonly used implant in daily clinical practice. Furthermore, there were no broad exclusion criteria, either regarding fracture types or patient characteristics. Second, we were not able to contribute clinical outcome data exceeding the primary hospital stay. This limits the overall significance of our study. Therefore, further outcome data are needed to effectively assess long-term clinical outcomes and any subsequent complications. During the aforementioned study period, we did not observe any implant-associated complications. Nonetheless, future studies are needed to prospectively assess perioperative and long-term clinical, functional, and patient-reported outcomes to adequately compare osteosynthesis systems.

## 5. Conclusions

The FNS is a highly effective fixation system for the surgical management of FNF. It allows for a significant reduction in the duration of surgery, thereby potentially reducing surgery-related risks and complications.

## Figures and Tables

**Figure 1 medicina-58-00352-f001:**
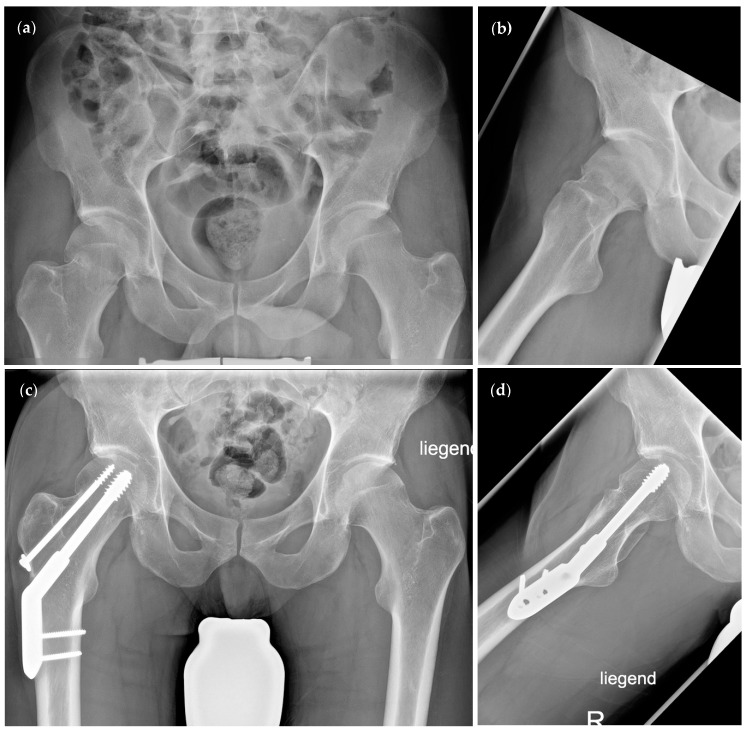
Radiographic visualisation of the dynamic hip screw (DHS) with anti-rotational screw (ARS). (**a**,**b**) Preoperative radiographs of a Garden type II/Pauwels type II femoral neck fracture. (**c**,**d**) Postoperative radiographs after osteosynthesis using a DHS with ARS.

**Figure 2 medicina-58-00352-f002:**
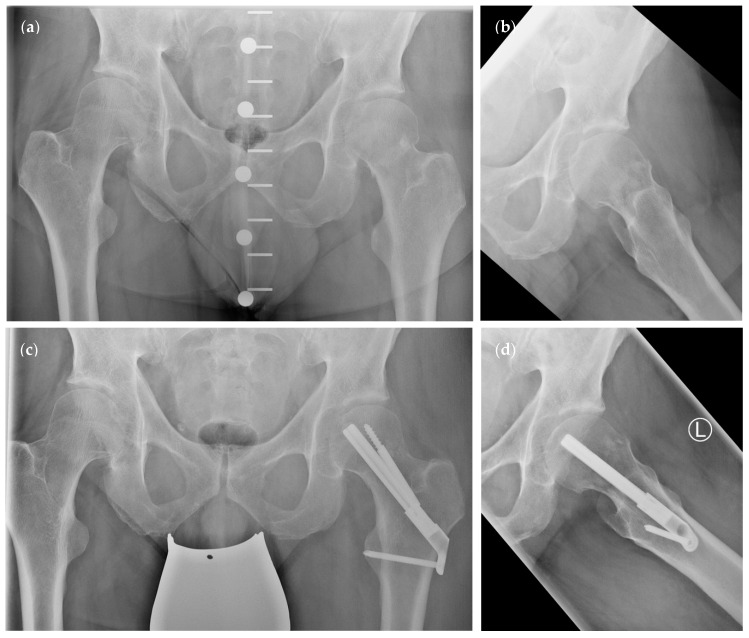
Radiographic visualisation of the femoral neck system (FNS). (**a**,**b**) Preoperative radiographs of a Garden type II/Pauwels type II femoral neck fracture. (**c**,**d**) Postoperative radiographs after osteosynthesis using the FNS.

**Figure 3 medicina-58-00352-f003:**
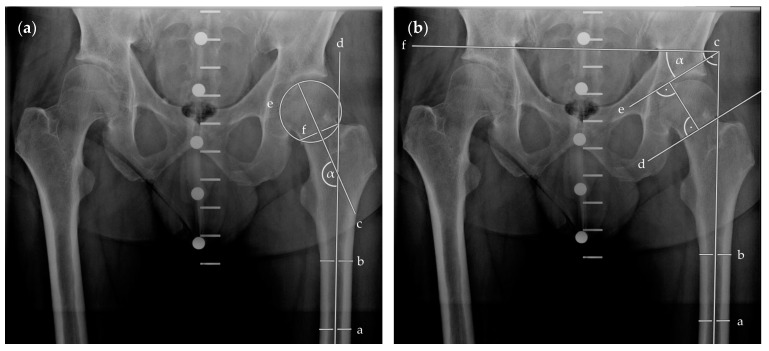
Radiographic visualisation of angle measurements in plain anterior–posterior radiographs of the pelvis. In (**a**), CCD angle (α) is measured between the longitudinal femoral shaft axis (d), determined by two bisections of the shaft (a, b), and the femoral neck axis (c), determined by the centre of the femoral head (centre of [e]) and its overlap with the femoral neck (f). In (**b**), fracture angle according to Pauwels classification (α) is measured between the fracture line (d or e) and the horizontal (f), which was perpendicular to the longitudinal femoral shaft axis (c), determined by two bisections of the shaft (a, b). Abbreviations: CCD, centrum–collum–diaphyseal.

**Figure 4 medicina-58-00352-f004:**
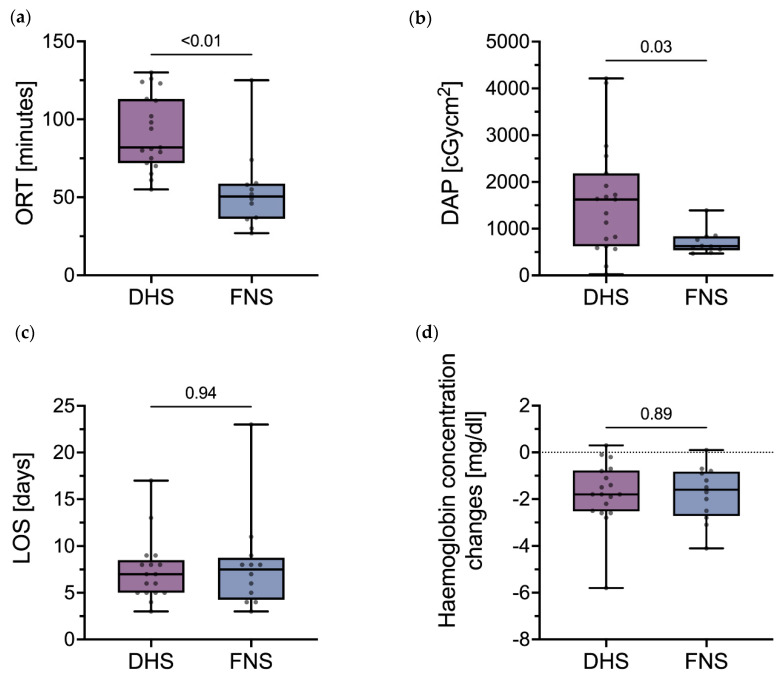
Outcome measures following osteosynthesis of FNF. (**a**) ORT, (**b**) DAP, (**c**) LOS, and (**d**) haemoglobin concentration changes in the DHS group and the FNS group. Abbreviations: FNF, femoral neck fracture; ORT, operating room time; DHS, dynamic hip screw; FNS, femoral neck system; DAP, dose-area product; LOS, length of stay.

**Table 1 medicina-58-00352-t001:** Overview of the study cohort.

		DHS (*n* = 19)	FNS (*n* = 12)	Statistics
Age (years)		60.47 ± 17	66.5 ± 10.98	*p* = 0.34
		(95% CI 52.28–68.67)	(95% CI 59.52–73.48)	
Gender	Female (% of group)	10 (52.63%)	6 (50%)	*p* = 1
	Male (% of group)	9 (47.37%)	6 (50%)	
ASA		2.32 ± 0.75	2.42 ± 0.67	*p* = 0.81
		(95% CI 1.96–2.68)	(95% CI 1.99–2.84)	
CCI		3.16 ± 3.39	4.42 ± 3.7	*p* = 0.38
		(95% CI 1.53–4.79)	(95% CI 2.06–6.77)	
Preoperative Pauwels angle (°)		50.93 ± 14.07	47.66 ± 14.44	*p* = 0.41
		(95% CI 44.15–57.71)	(95% CI 38.49–56.83)	
Postoperative Pauwels angle (°)		46.74 ± 7.71	43.34 ± 7.93	*p* = 0.22
		(95% CI 43.03–50.46)	(95% CI 38.3–48.38)	
Preoperative CCD angle (°)		129.5 ± 16.21	130.8 ± 13.25	*p* = 0.8
		(95% CI 121.7–137.3)	(95% CI 122.4–139.2)	
Postoperative CCD angle (°)		135.9 ± 7.27	136 ± 5.24	*p* = 0.85
		(95% CI 132.4–139.4)	(95% CI 132.7–139.4)	
Pauwels classification	Type I (% of group)	1 (5.26%)	1 (8.33%)	*p* = 0.72 *
	Type II (% of group)	10 (52.63%)	7 (58.33%)	
	Type III (% of group)	8 (42.11%)	4 (33.33%)	
Garden classification	Type I (% of group)	2 (10.53%)	1 (8.33%)	*p* = 0.45 **
	Type II (% of group)	9 (47.37%)	8 (66.67%)	
	Type III (% of group)	4 (21.05%)	2 (16.67%)	
	Type IV (% of group)	4 (21.05%)	1 (8.33%)	
TTS (min)		44.74 ± 10.66	48.83 ± 34.15	*p* = 0.16
		(95% CI 39.6–49.87)	(95% CI 27.13–70.53)	
In situ fixation	Yes (% of group)	2 (10.53%)	2 (16.67%)	*p* = 0.63
	No (% of group)	17 (89.47%)	10 (83.33%)	
Perioperative volume therapy (L)		1616 ± 661.2	1291 ± 784.2	*p* = 0.46
		(95% CI 1297–1934)	(95% CI 793.1–1790)	
Postoperative weight bearing	Partial weight bearing (% of group)	18 (94.74%)	10 (83.33%)	*p* = 0.54
	Full weight bearing (% of group)	1 (5.26%)	2 (16.67%)	
Discharge status	Stationary rehabilitation (% of group)	5 (26.32%)	7 (58.33%)	*p* = 0.13
	Home (% of group)	14 (73.68%)	5 (41.67%)	

Abbreviations: DHS, dynamic hip screw; FNS, femoral neck system; ASA, American Society of Anesthesiologists (physical status classification system); CCI, Charlson Comorbidity Index; TTS, time to surgery; CCD, centrum–collum–diaphyseal. * Fisher’s exact test assessing fracture distribution differences between groups (Type I + II vs. Type III). ** Fisher’s exact test assessing fracture distribution differences between groups (Type I + II vs. Type III + IV).

## Data Availability

The data presented in this study are available on request from the corresponding author. The data are not publicly available due to regulations of the local institutional ethics board.

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
