# Peer review of "Comparing Perioperative Outcome Measures of the Dynamic Hip Screw and the Femoral Neck System"

_medicina, 2022, doi:10.3390/medicina58030352_

Round 1

Reviewer 1 Report

This is a very interesting study about relative new femoral neck fracture fixation method (FNS) compared with DHS. In my opinion this study is suitable for publication after minor changes.

1., Chapter methods - e.g. complications should be mentioned (early, infection, revisions rate...)

2., Chapter discussion - add some sentence e.g: infection´s rate after DHS has been published about 1.3%, after FNS it was not until now widely puslished.

The citation - https://pubmed.ncbi.nlm.nih.gov/25105677/ should be included into the reference list

3., Chapter discussion - add please some sentence about this: the risc factors for the osteosynthesis failure, which should be considered before femoral neck fractures osteosynthesis (e.g. coxarthrosis, severe osteoporosis, rheumatism, chronic renal disease....) because the high risc of the DHS/FNS failure with the need for revision with THA. In some of these risky patient, primary THA should be considered.

Add this citation into the reference list: https://pubmed.ncbi.nlm.nih.gov/35172428/ 

Author Response

Dear reviewer 1,

thank you for your review of our submitted manuscript and for the potential improvements you pointed out. Please find our detailed answers to your recommendations down below.

Reviewer: “This is a very interesting study about relative new femoral neck fracture fixation method (FNS) compared with DHS. In my opinion this study is suitable for publication after minor changes. 1., Chapter methods - e.g. complications should be mentioned (early, infection, revisions rate...)”

Answer: Thank you for this hint. We have added this to our method’s section.

Reviewer: “2., Chapter discussion - add some sentence e.g: infection´s rate after DHS has been published about 1.3%, after FNS it was not until now widely puslished. The citation - https://pubmed.ncbi.nlm.nih.gov/25105677/ should be included into the reference list”

Answer: This is a very great supplementation of our discussion. We have added this fact concerning the infections following osteosynthesis using the DHS and included the mentioned reference

Reviewer. “3., Chapter discussion - add please some sentence about this: the risc factors for the osteosynthesis failure, which should be considered before femoral neck fractures osteosynthesis (e.g. coxarthrosis, severe osteoporosis, rheumatism, chronic renal disease....) because the high risc of the DHS/FNS failure with the need for revision with THA. In some of these risky patient, primary THA should be considered. Add this citation into the reference list: https://pubmed.ncbi.nlm.nih.gov/35172428/”

Answer: This is, as well, a very important note. We have added this to our discussion and included the reference. Thank you very much for you input.

Again, thank you for your thorough review. All of your constructive points significantly helped to improve our manuscript.

Best regards

The authors

Reviewer 2 Report

Dear Authors,

this is a well-written scientific paper. I do have only minor remarks and comments. See the reviewed version of your manuscript (PDF-File is attached).

It is clearly recognizable that the manuscript comes from German-speaking authors. I would recommend having it revised by a native-speaking editor so that it reads more fluently.

Author Response

Dear reviewer 2,

thank you for your review of our submitted manuscript and for the potential improvements you pointed out. Please find our detailed answers to your recommendations down below.

Reviewer: “Dear Authors, this is a well-written scientific paper. I do have only minor remarks and comments. See the reviewed version of your manuscript (PDF-File is attached).”

Answer: Thank you very much for your comprehensive review of our manuscript. We have worked on all recommendations that you included in the reviewed version of our manuscript and included the corrections in our revised manuscript. Further, we have added a figure to visualize the angle measurements

Reviewer: “It is clearly recognizable that the manuscript comes from German-speaking authors. I would recommend having it revised by a native-speaking editor so that it reads more fluently.”

Answer: Thank thank you for your assessment and your honesty. The manuscript has been revised by a native-speaking editor in order to improve legibility.

Again, thank you for your thorough review. All of your constructive points significantly helped to improve our manuscript.

Best regards

The authors